# Search-and-remove genome editing allows selection of cells by DNA sequence

Luise Fast [1], Madina Omar [1], Philipp Kanis [1], Theresa Schaffer [1], Deepika Chowdhury[1], Eva Rakava[1], Svante Pääbo [1,2] & Stephan Riesenberg [1] ✉

The selection of cells that have acquired a desired gene edit is often done by the introduction of additional genes that confer drug resistance or encode fluorophores. However, such marker genes can have unintended physiological effects and are not compatible with editing of single nucleotides. Here, we present SNIPE, a method that allows the marker-free selection of edited cells based on single nucleotide differences to unedited cells. SNIPE drastically enriches for cells, which have been precisely edited (median 7-fold). We validate the approach for 42 different edits using Cas9 or Cas12a in different cell types and species. We use it to enrich for combinations of substitutions that change missense mutations carried by all people today back to the ancestral state seen in Neandertals and Denisovans. We also show that it can be used to kill cultured tumor cells with aberrant genotypes and to repair heterozygous tumorigenic mutations.

Many methods have been developed to increase the efficiency with which precise substitutions and large knock-ins are introduced into genomes by editing[1–6]. Still, efficiencies can sometimes be very low for certain targets, especially for knock-ins of several kilobases. Low editing efficiencies can be compensated for by selection for cells carrying desired modifications. Generally, this is achieved by the introduction of a selectable gene encoding a fluorophore[7] or antibiotic resistance[8] or by the editing of an endogenous gene that conveys drug resistance[9,10] or alters fluorescence[11] simultaneously with the editing of the genomic feature of interest. However, these approaches can interfere with the physiology of the edited cells[12]. For example, antibiotic treatment can result in premature chain termination during translation, misfolded proteins, and selection of cells with chromosomal abnormalities[13,14]. Therefore, there is a need for selection methods that do not rely on the introduction of additional mutations or genes.

In CRISPR editing, a complex of a nuclease and a guide RNA (gRNA) can introduce a DNA double-strand break (DSB) at a site defined by the gRNA targeting sequence[15]. In yeast and bacteria, repeated CRISPR-assisted Cas9-induced DSBs have been shown to select for edited cells that have lost the gRNA recognition sequence[16–20]. Unfortunately, this has not been possible in higher eukaryotes, because they efficiently repair DSBs by non-homologous end joining (NHEJ) or microhomology-mediated end joining (MMEJ)[21,22]. Both DNA repair pathways can result in small insertions and deletions (indels) at the cut site that disrupt the gRNA recognition sequence and prevent re-cutting that would otherwise result in p53-dependent apoptosis[23]. Another DSB repair pathway is homologous recombination (HR) and the related homology-directed repair (HDR), which utilize homologous chromosomes or exogenous DNA templates for precise repair of the DSB. HR is less efficient than NHEJ or MMEJ in eukaryotic cells[24], but it is the predominant repair pathway in yeast and bacteria[24,25], where NHEJ and MMEJ are often absent[26–28]. Recently, inhibition of NHEJ and MMEJ has been achieved and used to increase HDR efficiency during editing with DNA donors in human cells[2,3,29].

Here, we present a marker free selection approach utilizing CRISPR gRNA-based cleavage of unmodified target DNA combined with inhibition of NHEJ and MMEJ. This allows the generation of cell populations made up of up to hundred percent edited cells based on DNA sequence differences that can be single nucleotides to the unedited cells.

## Results

### "SNIPE": selective nuclease-induced purity enhancement

Any mutation introduced by genome editing that destroys the gRNA recognition site renders the edited cells resistant to subsequent

[1]Max Planck Institute for Evolutionary Anthropology, Leipzig, Germany. [2]Human Evolutionary Genomics Unit, Okinawa Institute of Science and Technology, Onna, Japan. ✉e-mail: stephan_riesenberg@eva.mpg.de

cleavage, so that only unedited cells will experience DSBs. We set out to exploit the ability of gRNA to discriminate the target DNA of unedited and edited cells and combined this with the simultaneous inhibition of NHEJ (using M3814, a small molecule DNA-PKcs inhibitor) and MMEJ (by siRNA mix targeting *POLQ* mRNA)[2] to prevent indel formation and induce cell death in unedited cells (Fig. 1a). We dub this approach "selective nuclease-induced purity enhancement" or "SNIPE".

While knock-in of genes or changes of multiple bases can fully disrupt the gRNA recognition site, single-nucleotide changes can also prevent cleavage and are thus suited for the application of SNIPE. This is particularly the case for nucleotide changes located in the protospacer adjacent motif (PAM) or close to the PAM (Fig. 1b). For Cas9, changing a guanine in the 3′ NGG PAM sequence, or changing a single position in a core sequence located at +4 to +7 position upstream of

the PAM, strongly prevents cleavage (77–100%)[30]. For Cas12a (Cpf1), this is achieved by changing a thymine in the 5′-TTTN PAM sequence, or by changing a single position located at +2 to +7 position downstream of the PAM (78–100%)[31]. Single nucleotide mismatches at other positions in the gRNA recognition site only moderately prevent cleavage.

## Comparison of SNIPE and antibiotic selection

We compare SNIPE to state-of-the-art antibiotic selection, which requires the introduction of a drug resistance gene alongside the intended mutation during the editing experiment. In the next step, the cells are then treated with the appropriate antibiotic that allows only cells that have incorporated the DNA donor to survive (Fig. 1a). For this approach, puromycin is the drug of choice in mammalian cells[32–34]. We used prime editing (PE) in the 409B2 human induced pluripotent stem

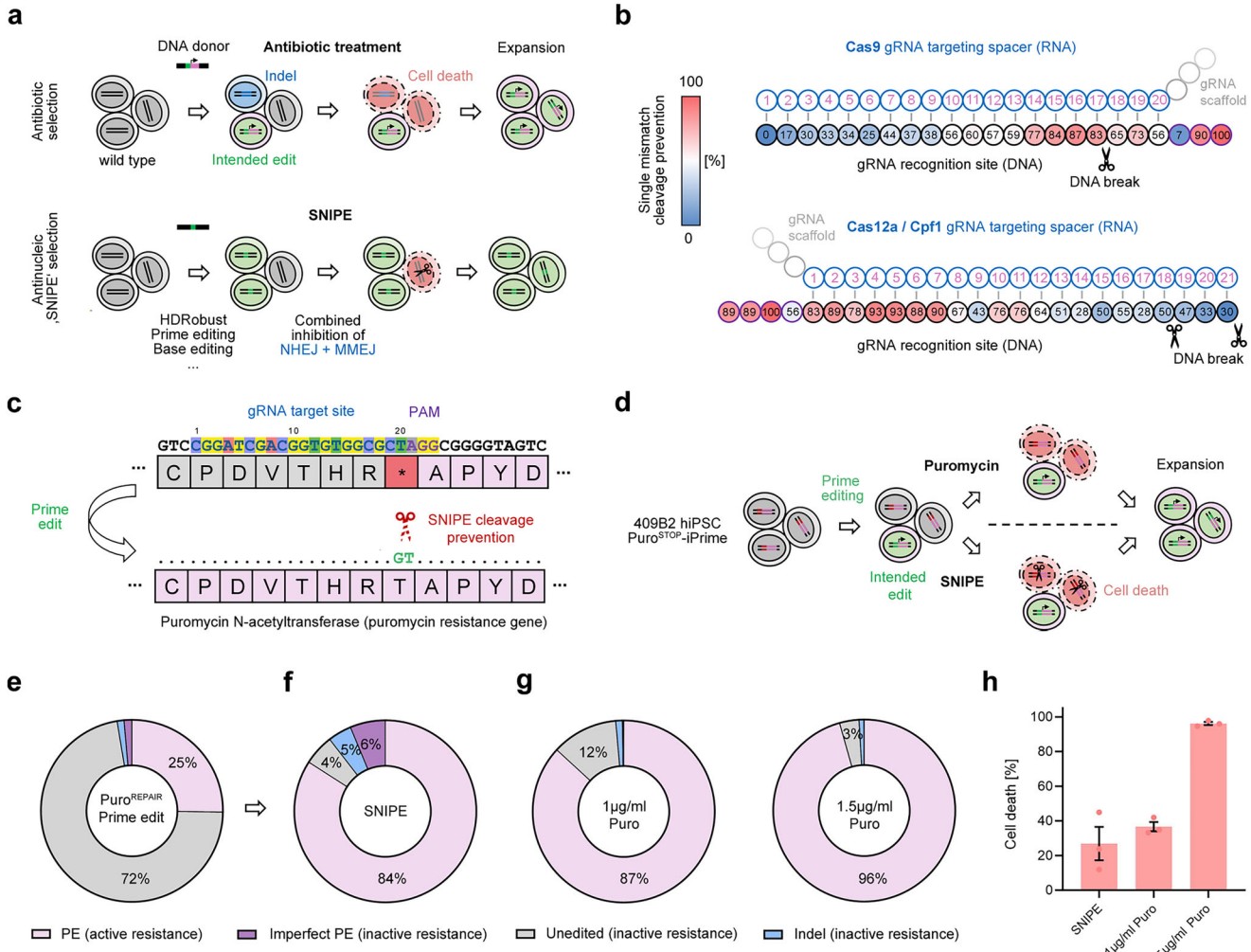

**Fig. 1 | Comparison of antibiotic selection and antinucleic "SNIPE". a** Strategy of antibiotic selection compared to antinucleic "SNIPE" selection. Unedited cells are colored in gray, cells with the intended edit in green and edited cells with indels in blue. Active antibiotic resistance is shown in rose and dying cells in red. **b** Strength of single-mismatch cleavage prevention by nucleotide distance from the PAM for Cas9[30] and Cas12a/Cpf1[31]. The percentage of cleavage prevention is shown as a color gradient from blue (weak prevention) to red (strong prevention). The base positions that are opposite to the PAM (NGG for Cas9, TTTV for Cas12a/Cpf1) on the non-target strand are indicated by purple frames. **c** Sketch of part of the puromycin resistance gene with a stop codon and surrounding amino acid sequence. Prime editing reverts the stop codon (depicted by red asterisk) to threonine which activates the puromycin resistance. The introduced edit destroys the gRNA target site and prevents SNIPE cleavage of the functional

puromycin resistance gene. **d** PE of 409B2 iPrime stem cells with a stop codon in the puromycin resistance gene can restore the puromycin resistance in successfully edited cells. The mixed cell population is then split for subsequent selection with puromycin or SNIPE. Unedited cells are colored in gray, cells with the intended edit in green, and edited cells with indels in blue. Active antibiotic resistance is shown in rose and dying cells in red. **e** Genome editing efficiencies for puromycin resistance by PE before selection. PE is depicted in rose, indels in blue, imperfect PE in purple and unedited cells in gray. **f** Genome editing efficiencies of cells from e selected by SNIPE. **g** Genome editing efficiencies of cells from e selected by 1 or 1.5 μg/ml puromycin. **h** Cell death after selection by SNIPE or antibiotic treatment quantified by a resazurin assay. Independent biological replicates were performed (*n* = 3) and are depicted as dots. Error bars show the s.e.m.

cell (hiPSC) line carrying a puromycin resistance gene with a premature stop codon that can be activated by editing of this stop codon (TAG) to a threonine codon (ACG) (Fig. 1c). PE is a widely used genome editing tool, in which a reverse transcriptase uses the nicked strand of the target site as a primer to introduce edits from PE gRNA (pegRNA)[35]. However, PE efficiency strongly varies among target sites, calling for enrichment of desired editing events.

We stably introduced an inducible PE system carrying a human codon optimized RT linked to iCRISPR–Cas9H840A (iPrime)[2] in the hiPSC line carrying the inactivated puromycin resistance gene. We then delivered pegRNAs that were designed to revert the stop codon to a threonine codon (Fig. 1d) and carried 5′- and 3′-end phosphorothioate bonds and 2′-OMe residues to prevent degradation[36]. We achieved 25% correction of the stop codon as assessed by Illumina sequencing of the target site (Fig. 1e). An additional 3% of editing events were due to imperfect PE, intended edits with additional unintended mutations, and indels. We then divided the cells for subsequent selection by SNIPE and by puromycin.

To enrich for edited cells by SNIPE, we electroporated a recombinant high fidelity Cas9 variant (Cas9-HiFi)[37] complexed with Cas9 SNIPE-gRNA targeting the sequence with the puromycin stop codon as ribonucleoprotein (RNP) while simultaneously inhibiting NHEJ and MMEJ[2]. This resulted in that 84% of the cells carried the correction of the stop codon (3.2-fold increase) (Fig. 1f). Due to the presence of imperfect PE and indels before selection, SNIPE also increased the proportion of unintended editing events to 10%. We then generated 41 single-cell-derived cellular clones to assess the extent of large deletions or rearrangements after SNIPE and found no evidence for unintended copy number changes of the target locus (Supplementary Fig. 1).

Selection using 1 µg/ml puromycin for 3 days rather than by SNIPE resulted in 87% cells carrying activated puromycin resistance, comparable to SNIPE (Fig. 1g). SNIPE resulted in the death of 27% of unedited cells as estimated by a resazurin assay 3 days after editing. A comparable percentage of cell death of 37% was observed for 1 µg/ml puromycin treatment (Fig. 1h). Stop codon correction could be further increased to 96% by applying 1.5 µg/ml puromycin, but at the cost of very high cell death (96%) due to puromycin toxicity (Fig. 1g, h).

## Selecting for intended point mutations

In addition to the above tested inhibition of NHEJ and MMEJ by M3814 and *POLQ* siRNAmix, we next compared the performance of SNIPE using other combinations of substances for end-joining pathway inhibition: M3814 alone, M3814 and PolQi2 (PolΘ inhibitor), AZD7648 (DNA-PKcs inhibitor) and PolQi2[3,38]. We targeted the gene *FANCF* with PE (+5G to T) and obtained 20% iPrime PE efficiency (Fig. 2a). SNIPE using M3814 and *POLQ* siRNAmix, M3814 and PolQi2, AZD7648 and PolQi2 resulted in 73%, 72%, and 62% PE efficiency, respectively (Supplementary Fig. 2). M3814 alone increased efficiency to 70%. We and others have observed an increased tendency for unintended large-scale on-target effects when NHEJ is inhibited[2,3,39]. We therefore isolated cellular clones derived from single cells edited in the presence of the different substance combinations, sequenced the target locus and used droplet digital (dd) PCR to determine the copy number, enabling the detection of deletions, duplications, translocations, and chromosome losses. SNIPE using M3814 and *POLQ* siRNAmix resulted in 2.5% clones with copy number changes, while dual end-joining inhibition with two small molecules resulted in 11–13%. As expected, sole NHEJ inhibition by M3814 had the highest number of clones with copy number changes (21%).

After having established that M3814 and *POLQ* siRNAmix is the best formulation for SNIPE, we tested additional targets by PE in the genes *VEGFA* (+5G to T) and *HEK3* (+1T to A) for which pegRNAs have been optimized[35]. We obtained PE efficiencies of 3% for *VEGFA* and 8% for *HEK3*. The outcome purities, i.e., the proportion of editing events

that were the intended ones, were 69–80% (Fig. 2a). SNIPE increased PE outcomes to 22% for *VEGFA*, 17% for *HEK3* and 77% for *FANCF* (mean 4.2-fold) and increased purities to 81–95%. Re-cutting the targets in the edited bulks using Cas9-HiFi RNP without inhibition of the two end-joining pathways moderately increased outcomes for two out of three targets (mean 2-fold), but drastically reduced purity to a mean of 25%. In addition to PE, we also employed base editing using recombinant ABE8e[40] and could increase base editing efficiency from 23% to 53% at the *AHR* target site using SNIPE (Supplementary Fig. 3).

To assess how SNIPE performs for heterozygous or homozygous genotypes, we used PE to edit *FANCF* (+5G to T) and isolated single cell-derived clones with the genotypes: WT/WT (wild type), PE/PE, PE/WT, and PE/indel (Supplementary Fig. 4). Subsequent SNIPE targeting the wildtype allele drastically killed both WT/WT (99% cell death) and PE/WT (98%), but hardly killed PE/PE (9%) and PE/indel (0%). Interestingly, SNIPE of the PE/WT clone increased PE sequencing reads from 50% to 75%, suggesting that in the few surviving cells, DSBs introduced by SNIPE were repaired by HR using the PE allele as a template. Consequently, SNIPE enriches for biallelic editing and can remove cells carrying at least one wild type allele from the cell population.

Next, we edited three targets where HDR efficiencies are low: *CDH16* (T342A), *AHR* (V381A), and *PRDM10* (N1129T). We used Cas9, gRNA, and exogenous single-stranded (ss) DNA donors to introduce the missense mutations with or without dual end-joining repair pathway inhibition (HDRobust) (Fig. 2b). We also tested SNIPE with GOLD-gRNAs, which we have previously developed to increase cleavage efficiency[41]. HDRobust increased HDR efficiency and purity for all targets (from 9–32% to 81–84% purity) relative to standard HDR editing with a DNA donor. SNIPE further increased HDRobust outcomes to 39% for *CDH16*, 33% for *AHR* and 49% for *PRDM10* and increased purities (89–93%). Efficiencies were even better with GOLD-SNIPE (49–56%). We furthermore tested two targets where HDR efficiencies are high, *C3* (A1286V) and *DCHS1* (D777N). We achieved 84% and 94% HDR when using HDRobust followed by GOLD-SNIPE (Fig. 2c).

As expected, SNIPE is compatible with Cas12a editing since we could enrich desired edits after HDRobust from 20–59% to 43–91% for editing of *NCOA6* (I823M), *IZUMO4* (R185r.185_197del), and *GLDC* (L220F) with Cpf1-Ultra[42] RNP (Fig. 2d). We also used SNIPE in a chimpanzee iPSC line, using M3814 to inhibit NHEJ and a siRNA designed to target the chimpanzee *POLQ* to inhibit MMEJ. We achieved 80% precise editing of *DCHS1* with SNIPE (Fig. 2e).

Across the tested targets, SNIPE increases the median editing efficiency 7-fold compared to normal editing and, surprisingly, also increases outcome purity (Fig. 2f). The increase in purity is likely due to that SNIPE can target editing-induced indels that do not destroy the gRNA recognition site. Indeed, SNIPE often alters the indel pattern generated in the initial editing experiment (Supplementary Fig. 5).

To also analyze cellular clones after SNIPE in chimpanzee cells, we generated 24 single-cell-derived clones from the chimpanzee *DCHS1* edit and quantified the number of homozygous and heterozygous edits. 79% of the precisely edited clones were homozygous, indicating a preferential selection of biallelic intended edits by SNIPE (Supplementary Fig. 6). Four clones (17%) have co-converted a SNP 52 bp downstream from the intended edit, but none showed copy number loss. To assess off-target editing, we scored the editing efficiency of the two most likely predicted off-target sites for each of the targets for which Cas9 and Cas12a were used in this study[43,44]. Mean off-target editing with standard editing or SNIPE was 2.7% (0–47.2%) or 0.2% (0–2.1%), respectively. The *CDH16* edit, which stood out with 47.2% combined off-targets with standard editing, was reduced to 2.1% with SNIPE (22-fold) (Supplementary Figs. 7 and 8).

## Multiplexed enrichment of single-nucleotide edits

Multiplexed simultaneous precise editing in 409B2 hiPSCs using CRISPR enzyme RNP and HDRobust results in that several DNA DSBs

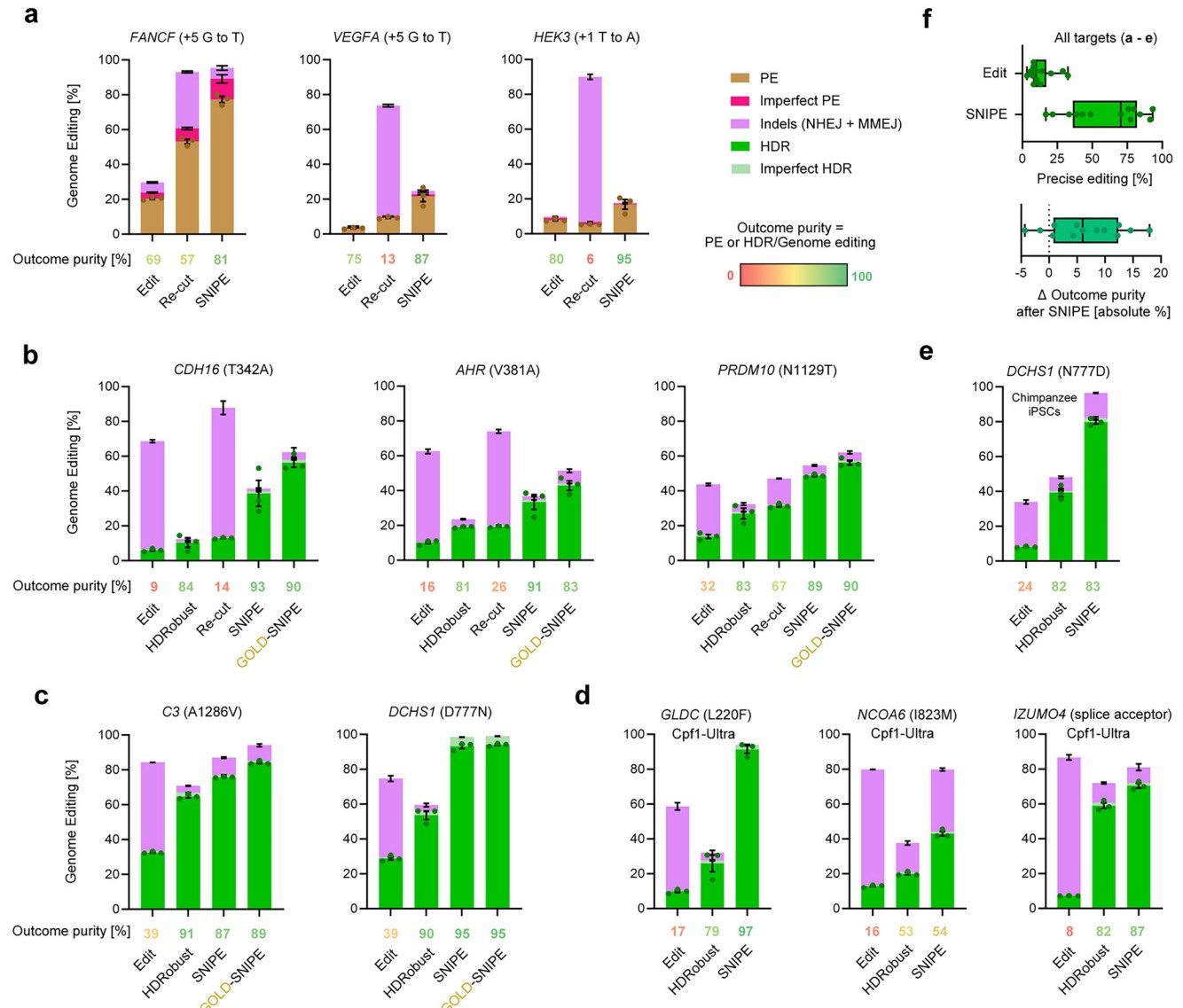

**Fig. 2 | SNIPE efficiencies across targets, CRISPR enzymes and cell lines.**
**a** Genome editing efficiencies for prime editing of human 409B2 iPrime stem cells of different targets before and after re-cutting (without repair pathway inhibition) or SNIPE (with repair pathway inhibition). The mean outcome purity (percentage of precise intended editing relative to all editing events) is given below the graphs. PE is indicated in brown, imperfect PE in light brown, and indels in magenta. **b** Genome editing efficiencies for low HDR efficiency Cas9 targets without pathway inhibition (edit), with pathway inhibition (HDRobust), and subsequent re-cutting without repair pathway inhibition (Re-cut), SNIPE, and GOLD-SNIPE. GOLD-SNIPE uses GOLD-gRNAs engineered for more efficient cleavage[41]. HDR is indicated in green, imperfect HDR in light green, and indels in magenta. **c** Editing and SNIPE for high HDR efficiency targets with Cas9. **d** Editing and SNIPE with Cas12a (Cpf1-Ultra). **e** Editing and SNIPE in Sandra A chimpanzee induced pluripotent stem cells. **f** Precise editing before and after SNIPE and change in outcome purity after SNIPE across all targets shown in (**a**–**e**). Independent biological replicates were performed (n = 3 for **a**–**e**) and are depicted as dots. Error bars show the s.e.m. In **f** each dot indicates the mean of one target, boxes the 25th to 75th percentile, lines medians and whiskers extend from minimum to maximum values.

are generated at the same time, and therefore in low editing efficiencies even for high-efficiency targets. Nevertheless, SNIPE alone, and even more so GOLD-SNIPE drastically increases the percentage of chromosomes in a cell population, which carry three desired edits from 1.3–3.5% to 23–46% (mean 16-fold) (Fig. 3a).

Motivated by the potential of precision editing and subsequent SNIPE to change many positions in the same cell, we decided to introduce a combination of five missense mutations in five genes back to the ancestral state not observed to date in any present-day human but observed in all Neandertal and Denisovan genomes available[45,46]. These changes have thus occurred and reached very high frequencies during the past half million years of human evolution. To avoid DSBs and genomic instability, we used PE and designed pegRNAs and respective nicking gRNAs, which only rarely introduce DSBs. After

expression of the prime editor in the 409B2 iPrime stem cell line, we introduced the five pegRNA/nicking gRNA pairs by electroporation. This resulted in editing efficiencies of 5–12% for *IZUMO4* (R185r.185_197del), *CDH16* (T342A), and *PROM2* (D458E), while no editing was observed for *AHR* (V381A) and *NCOA6* (I823M) (<0.02%) (Fig. 3b). Three additional sequential PE rounds for the former three genes increased the percentage of edited chromosomes in the mixed cell population to 24% for *IZUMO4*, 30% for *CDH16*, and 57% for *PROM2* (Fig. 3c). Subsequent single HDRobust edits for *AHR* and *NCOA6* on this mixed cell population resulted in precise editing of 45% and 26%, respectively. We then applied SNIPE of one target, two targets, and three targets of the three genes with the lowest editing frequencies of precise edits to increase the percentage of cells precisely edited for all genes (Fig. 3d). After these experiments and assuming that the edits

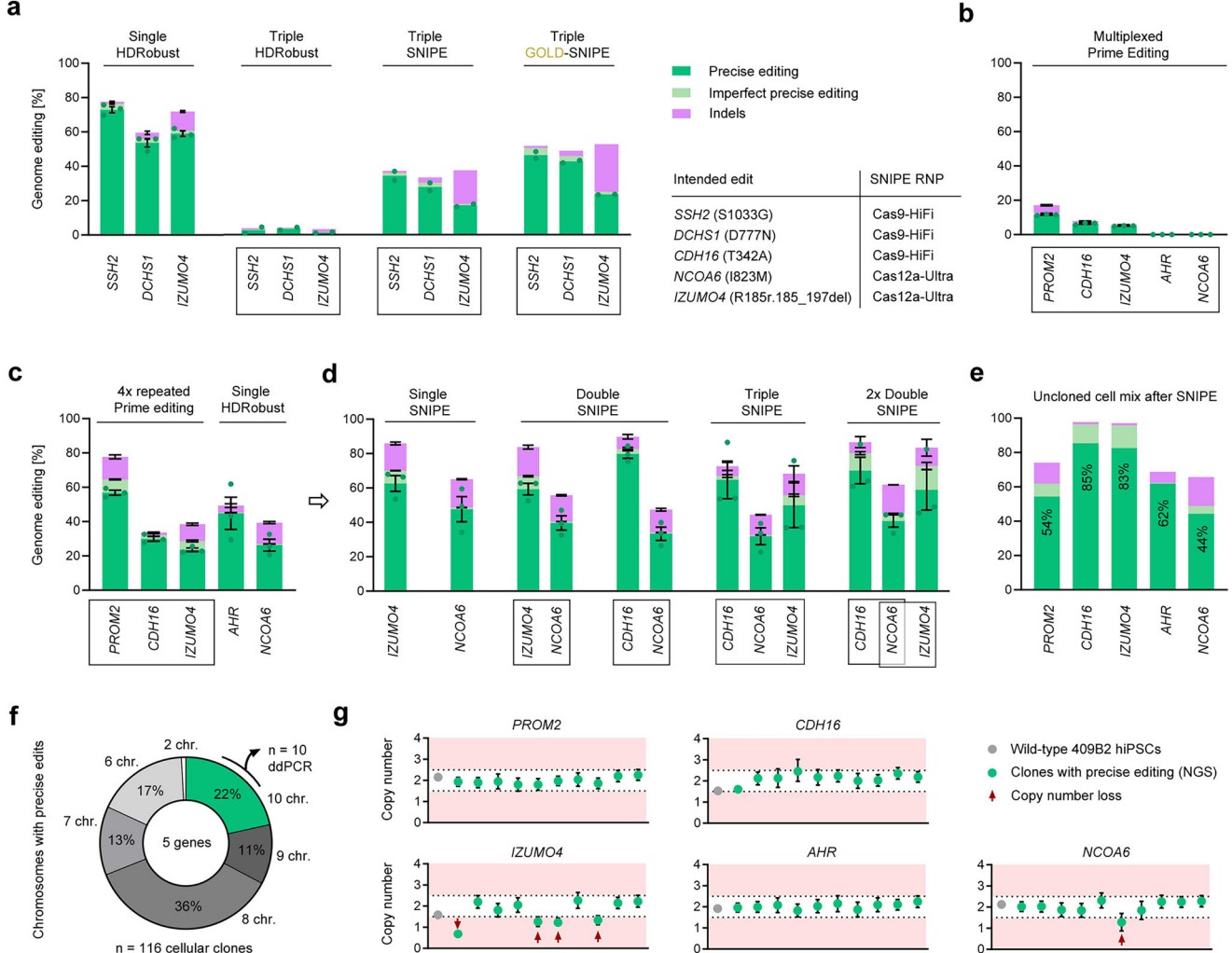

**Fig. 3 | Multiplexed enrichment of point mutations by SNIPE. a** Comparison of single and triple HDRobust edits for *SSH2, DCHS1* and *IZUMO4* as well as subsequent SNIPE or GOLD-SNIPE in 409B2 stem cells. Precise editing is indicated in green, imperfect precise editing in light green and indels in magenta. **b** Genome editing efficiencies for multiplexed prime editing (PE) in the 409B2 iPrime stem cell line for five genes. **c** Genome editing efficiencies after four rounds of repeated multiplexed PE for *PROM2, CDH16* and *IZUMO4* as well as subsequent single HDRobust editing for *AHR* and *NCOA6*. **d** Genome editing efficiencies after single, double, and triple SNIPE. **e** Genome editing efficiencies for all five genes in the uncloned cell mix with the highest precise editing efficiencies after SNIPE. **f** Percentage of precisely edited chromosomes for five genes in 116 cellular clones. Clones with precise edits on all 10 targeted chromosomes are highlighted in green. **g** Copy number analysis by

droplet digital PCR of ten cellular clones edited for all ten edited chromosomes for the five target genes based on NGS. The copy number of target sequences relative to the gene *FOXP2* in cellular clones is plotted as a green circle, while the wild type control is shown in gray. The measure of center for the error bars represents the ratio of the Poisson-corrected number of target to reference molecules multiplied by two for the diploid state of the reference gene. The error bars represent the 95% confidence interval of this measurement. Cellular clones with apparent copy number loss are marked by a red arrow. Independent biological replicates were performed (*n* = 3, exception *n* = 2 for triple edits in a) and are depicted as dots for precise editing. Error bars show the s.e.m. Enzymes used for SNIPE for the different targets are indicated in a table.

occur independently of each other, the best cell pool had an estimated 10% of cells carrying all five edits (Fig. 3e). From this pool, we sorted single cells to generate clones. Of the 116 cellular clones, 25 (22%) carried the desired five edits on all 10 chromosomes (Fig. 3f). We then used dd PCR to estimate the copy numbers of the five edited targets in the cellular clones. Four clones had an apparent copy number loss in the *IZUMO4* target, and of these clones, one had an additional copy number loss in *NCOA6* while the other six targets had two copies of all five targets (Fig. 3g). In addition, a previously heterozygous SNP 21 kb downstream of the *IZUMO4* target had become homozygous in all ten clones, suggesting it had been converted by the sister chromatid in the process of editing and SNIPE. One out of ten clones also became homozygous for positions flanking the *NCOA6* target site, while no loss of heterozygosity was observed for SNPs in the three other targets (Supplementary Fig. 9).

## Selecting for large knock-ins

Precise knock-ins of large DNA fragments or entire genes remain often inefficient in most cell types, especially iPSCs[47]. We therefore set out to test the ability of HDRobust in combination with with double-stranded (ds) DNA donors and SNIPE to increase knock-ins. We electroporated 409B2 hiPSCs carrying a stably introduced doxycycline inducible expression system in the AAVS1 safe harbor locus[48] with Cas9-HiFi RNP and a 1.4 kb dsDNA donor, encoding the transcription factor ETV2[49] (Fig. 4a, b). After editing with or without HDRobust, we genotyped a total of 178 single-cell-derived clones by gel electrophoresis of PCR amplification products of the target locus (Fig. 4c). Without repair pathway inhibition, 0.9% of clones carried the *ETV2* insert, while 16% did so after application of HDRobust.

We then applied SNIPE to generate 19 cell lines carrying cDNA expression constructs encoding transcription factors (TF) or TF

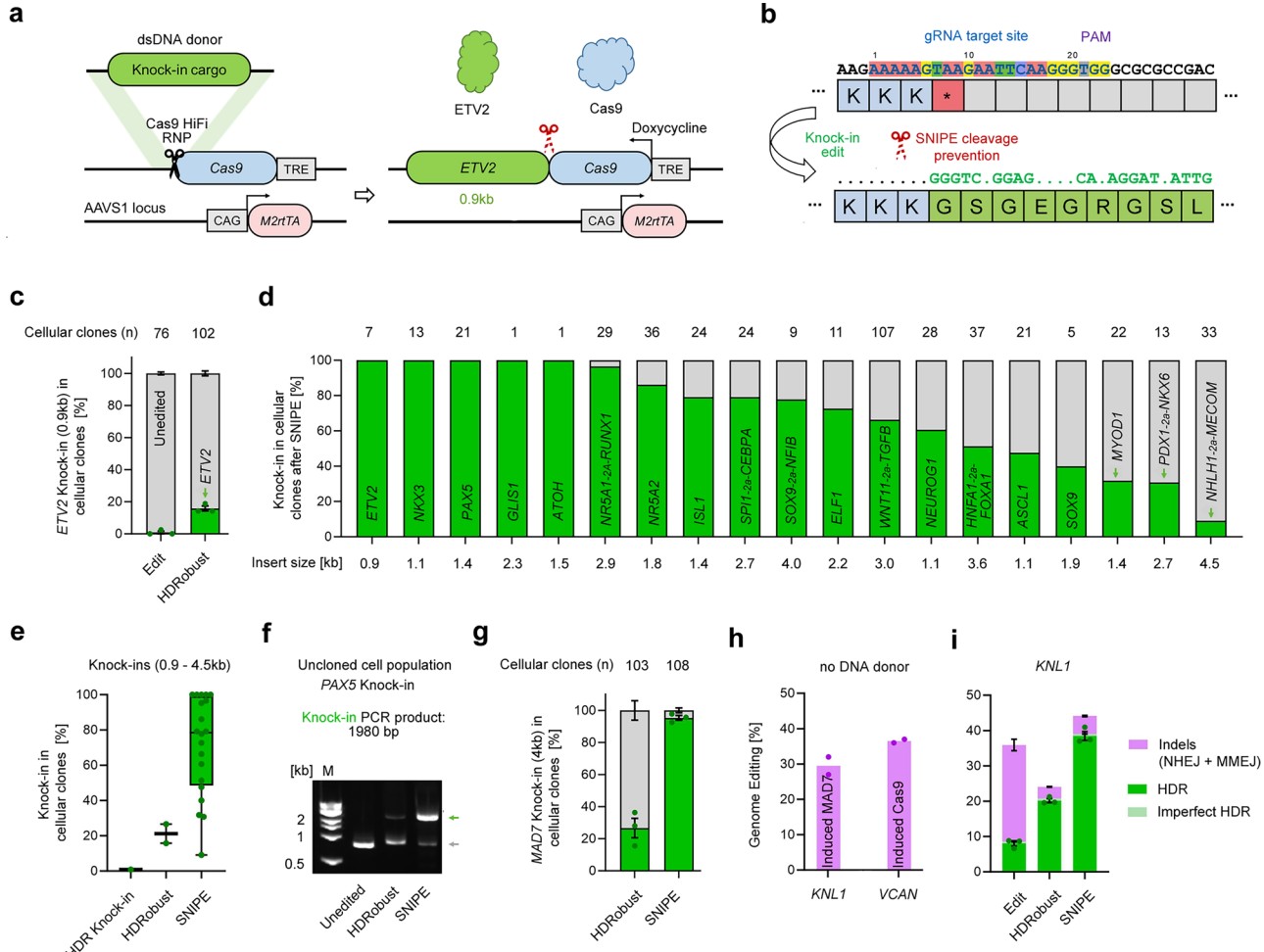

**Fig. 4 | Selection of gene knock-ins by SNIPE. a** Strategy for the knock-in of a dsDNA donor behind the Cas9 gene in the iCRISPR AAVS1 safe harbor locus in 409B2 stem cells by editing with a Cas9 Hifi RNP. **b** Insertion of *ETV2* (green) in the iCRISPR cassette changes the gRNA target site and prevents cleavage by SNIPE. **c** Knock-in efficiency of *ETV2* without (edit) and with pathway inhibition (HDRobust). Clones were genotyped by PCR amplification of the insert and gel electrophoresis. Knock-in clones are depicted in green and unedited clones in gray. Each dot shows the mean percentage of knock-in clones after single-cell sorting after independent editing (*n* = 3). **d** Knock-in efficiencies of 19 transcription factors or transcription factor combinations. Numbers of cellular clones are indicated above and insert size in kb below. Inducible transcription factor (iTF) knock-in clones are colored in green and wild type clones in gray. **e** Distribution of knock-in efficiency across all analyzed large knock-ins and conditions in (**c**), (**d**), (**g**). Each dot indicates the mean of one target

(*n* = 1 HDR knock-in, *n* = 2 HDRobust, *n* = 20 SNIPE), boxes the 25th to 75th percentile, lines medians and whiskers extend from minimum to maximum values. **f** Agarose gel of PCR products of uncloned cell populations shows unedited cells (lower gel band) and *PAX5* knock-in (upper gel band) with HDRobust before and after SNIPE. Agarose gels of PCR products showing expected knock-in bands as shown in f were independently repeated for *n* = 20 knock-ins. **g** Knock-in efficiency of *MAD7* with pathway inhibition (HDRobust) and subsequent SNIPE. Analysis was done as described for (**c**). **h** Genome editing efficiency for targeting *KNL1* and *VCAN* without a DNA donor by inducible MAD7 and Cas9, respectively. **i** Genome editing efficiency of *KNL1* editing with a DNA donor by MAD7 without (edit), with pathway inhibition (HDRobust), and subsequent SNIPE. HDR is indicated in green, imperfect HDR in light green and indels in magenta. Data are shown as mean of independent biological replicates (*n* = 3 for **c**, **g**, **i**; *n* = 2 for **h**) depicted by dots and error bars show the s.e.m.

combinations described to be able to directly generate various cell types[50,51] (Fig. 4d). The TF inserts ranged in size from 0.9 to 4.5 kb. The mean knock-in efficiency after SNIPE, determined by PCR of the target loci from single cell-derived clones across the TF inserts, was 70% (s.e.m. 6.5%). For five transcription factors, all of the generated clones carried the desired knock-in. Across all tested knock-in edits, SNIPE increased median efficiency 3.4-fold compared to HDRobust (Fig. 4e). This strong enrichment can easily be seen when comparing the intensity of PCR products from cell populations before cloning that had not been edited, edited with HDRobust, and enriched with SNIPE, respectively (Fig. 4f).

We furthermore generated a hiPCS line that can express both Cas9 and MAD7 (ErCas12a)[52,53] in order to be able to edit genomic sites adjacent to both NGG and TTTN PAMs[54]. Introduction of *MAD7* in a *Cas9*-containing cell line by HDRobust followed by SNIPE resulted in 95% of single cell-derived clones carrying a 4 kb DNA fragment encoding MAD7

(Fig. 4g). We confirmed that both Cas9 and MAD7 function in these cells by using gRNAs designed for two different targets and observing ≥30% indels when Cas9 and MAD7 were used (Fig. 4h). We also used induced MAD7 to edit a site in the gene *KNL1* (S1086G) followed by subsequent SNIPE with MAD7. Standard editing with a ssDNA donor resulted in 8% precise editing by HDR. Editing with HDRobust and subsequent SNIPE increased efficiency to 20% and to 38%, respectively (Fig. 4i).

**Selective killing of cultured cells with cancer mutations by SNIPE**

Since SNIPE can enrich genetically modified cells by killing unmodified cells, it should be possible to selectively kill cells carrying any genetic mutation, while sparing wild type cells, in a mixed cell population (Fig. 5a). To test this, we generated a 409B2 cellular clone carrying a biallelic stop codon in *TP53* (R72*) to model p53-deficient tumor cells[55]. We showed that the p53-deficient cells proliferate faster than wild type

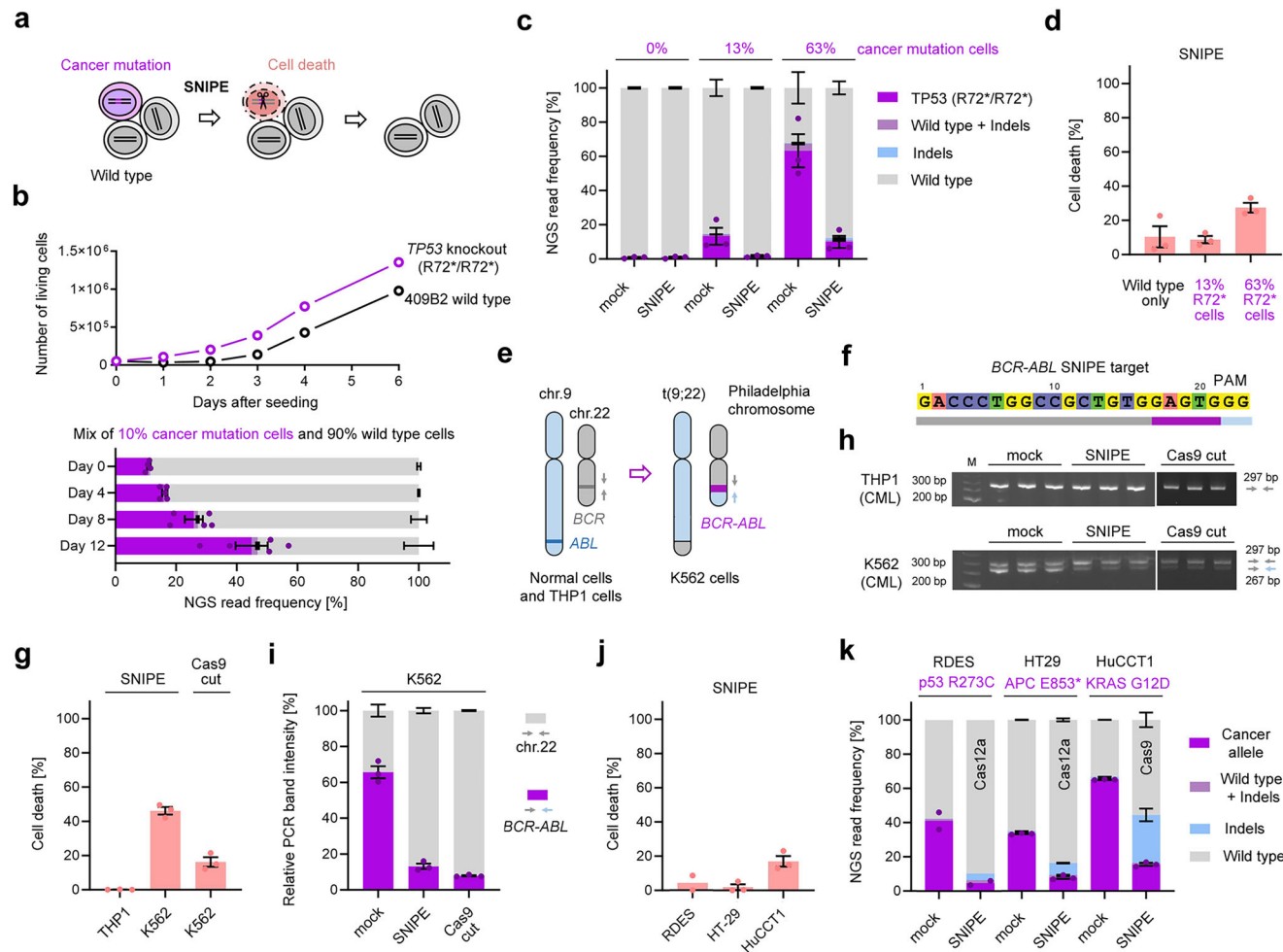

**Fig. 5 | Selective killing of cultured cells with cancer mutations by SNIPE.**
**a** Strategy to apply SNIPE to kill cancer cells based on cancer specific mutations. Wild type cells are colored in gray, cells with cancer mutations in purple and dying cells in red. **b** Cell growth curves of human 409B2 iPSCs measured by resazurin assay (upper panel). The lower panel shows NGS read frequency for cancer mutation (purple) and wild type cells (gray) in a mixed population over time. **c** Killing of *TP53* knock-out 409B2 cells by SNIPE. Different amounts of *TP53* knock-out cells (purple) were mixed with wild type cells (gray). Indels are colored in blue and wild type with indels in light purple. **d** Cell survival after selection by SNIPE related to (**c**). **e** K562 cells carry a Philadelphia chromosome, characterized by a reciprocal translocation of chromosome 9 and 22 resulting in the fusion oncogene *BCR-ABL1*. **f** The *BCR-ABL* fusion generates a new gRNA recognition site in K562 cells. **g** Cell survival of THP-1 and K562 cells after targeting the *BCR-ABL* junction. The label "Cas9 cut" indicates CRISPR targeting without SNIPE. **h** Agarose gel of PCR products of healthy chromosome 22 (297 bp, gray) and Philadelphia chromosome (267 bp, purple). **i** Quantification of band intensity for K562 cells of (**h**). **j** Cell survival of heterozygous cancer cell lines RD-ES, HT-29 and HuCC-T1 after targeting the disease allele with SNIPE. **k** NGS read frequency of cancer locus alleles in RD-ES, HT-29 and HuCC-T1 for mock and SNIPE edit. The cancer mutation allele is colored in purple, wild type in grey, indels in blue and wild type with indels in light purple. Independent biological replicates were performed (*n* = 3 for **b, c, d, g, h, i, j, k**, *n* = 2 for RDES in **j** and **k**) and are depicted as dots for the cancer mutation allele sequence reads and cell survival. Error bars show the s.e.m.

cells (Fig. 5b) so that they become more frequent in a mixed culture over time. We then used SNIPE to select against cells carrying the p53 mutation in cell populations where an average of 13% and 63% carried the *TP53* (R72*) mutation. The *TP53* (R72*) cells decreased in frequency from 63% to 10% (27% cell death) and from 13% to 2% (9% cell death), respectively (Fig. 5c, d).

We tested the ability of SNIPE to kill chronic myeloid leukemia (CML) K562 cells that carry a Philadelphia chromosome, i.e., a reciprocal translocation of chromosome 9 and 22 that results in the fusion oncogene *BCR-ABL1*[56,57] (Fig. 5e). This translocation generates a novel gRNA recognition site in K562 cells that is not present in normal cells (Fig. 5f). We targeted the *BCR-ABL1* junction in K562 cells with SNIPE, and used THP-1 CML cells, where the translocation is absent, as a control. SNIPE did not kill THP-1 cells, but killed approximately half of the K562 cells (Fig. 5g). CRISPR-targeting the *BCR-ABL1* locus without SNIPE killed 16% of cells, which is consistent with the toxicity resulting from the creation of multiple DNA double-strand breaks when targeting *BCR-ABL1* in K562 cells[58].

We used PCR amplification by primer pairs that amplify either 297 bp of healthy chromosome 22 or 267 bp of the Philadelphia chromosome to gauge the proportion of cells carrying the Philadelphia chromosome. This suggests a reduction of *BCR-ABL1* containing chromosomes from 66% to 13% among the surviving K562 cells, which is comparable to the reduction seen with standard CRISPR-targeting (Fig. 5h, i).

Finally, we investigated the impact of SNIPE on cell survival and genotype after targeting disease alleles in the heterozygous cancer cell lines RD-ES (p53 R273C), HT-29 (APC E853*), and HuCC-T1 (KRAS G12D). Only 2–17% cells died after SNIPE (Fig. 5j) while surprisingly disease mutations were reduced from 41% to 5% for RD-ES, 33% to 8% for HT-29 and 65% to 15% for HuCC-T1 and wild type alleles increased from 58% to 90%, 65% to 84% and 33% to 56%, respectively (Fig. 5k). Similar to our observations from Supplementary Fig. 4, we hypothesize that the DSBs introduced in the disease alleles by SNIPE were repaired by HR using the wild type chromosome as a template, resulting in the creation of cells without cancer mutations.

## Discussion

We have recently shown that a combination of a small molecule and siRNA can almost completely inhibit NHEJ and MMEJ, respectively, and that this drastically increases HDR efficiency and editing with exogenous donors (HDRobust)[2]. Here, we combine inhibition of NHEJ and MMEJ with selective CRISPR cleavage via gRNA mismatch discrimination of as little as one nucleotide to select for cells of desired genotypes (Fig. 1). This approach, "selective nuclease-induced purity enhancement" or "SNIPE", increases the median frequency of point mutations in editing experiments 7-fold (Fig. 2), can be used to select for multiple edited targets (Fig. 3) and for kilobase knock-ins of up to 4.5 kb (Fig. 4). Our data suggest that SNIPE can also prevent unintended on-target editing events when clones are analyzed, as well as off-target editing events at predicted sites (Supplementary Fig. 1, 2, 6, 8). To improve quality control and increase sensitivity in the detection of unintended side effects in pooled cell populations, profiling using CAST-seq[59] or low-pass whole genome sequencing for copy number variants and aneuploidy could be employed. Additionally, unbiased genome-wide off-target assessment[60,61] is necessary to exclude off-target editing events more rigorously. In contrast to editing, SNIPE does not use donor DNAs. This reduces the risk of activation of the p53-mediated DNA damage response, cellular senescence and inflammatory programs[62,63].

The combination of beneficial features of SNIPE makes it possible to generate populations of cells where the vast majority of cells carry the desired genotype. For many applications, this circumvents the need to generate clones from single cells and therefore avoids clone-to-clone variation that often complicates analyses of edited cells[64], and allows for faster phenotypic readouts, for example accelerated antibody development[65].

SNIPE requires the highly efficient delivery of potent ribonucleoproteins, which can be achieved for most common cell lines, including stem cells. However, CRISPR components are sometimes delivered encoded in plasmids that can be modified easily beforehand, but the delivery of plasmids is typically less efficient. For approaches with inefficient delivery, SNIPE should be combined with other methods developed to enrich for cells with successfully delivered CRISPR components[66,67] or cleavage activity on a plasmid surrogate reporter[68]. SNIPE is optimized in cell lines amenable to clonal derivation, but has not been validated in primary cells. The sequential delivery process may be less tolerated by such cell types, for which SEED[69] or SLEEK[70] selection might be suitable alternatives, as they can achieve efficiencies of more than 90%.

Optimal SNIPE gRNAs allow almost complete selection, but differences in efficiencies among targets suggest that cleavage efficiency of gRNAs is the main factor limiting the performance of SNIPE. To improve SNIPE performance with Cas9, GOLD-gRNAs with a highly stable RNA-hairpin can be used[41] (Fig. 2) as well as other approaches to increase the efficiency of Cas9 gRNAs[71,72] and Cas12a variant gRNAs[73,74]. SNIPE gRNAs should be designed so that the edited nucleotide produces a mismatch within the PAM or the seed region of the gRNA to confer optimal cleavage prevention[15,30,31,75,76]. Retargeting of the most common indel sequences by matching gRNAs, so called recursive editing, is another straightforward approach to increase precise genome editing while also reducing indels[77–79]. This could potentially be applied subsequently to SNIPE to further increase outcome purity. Because of the observations that SNIPE strongly enriches for biallelic editing (Supplementary Fig. 4), and that cells after PE and base editing are often only edited on one chromosome when the efficiencies in the cell bulk are low to moderate[80,81], biallelic fold-enrichment by SNIPE is likely much higher than the measured fold-increase of precise editing efficiency in the cell mix.

In addition to gRNA cleavage efficiency that can limit the success of editing, a potential problem are large scale on-target effects that cannot be quantified by standard sequencing after PCR amplification of the target site. It is promising that frequent, large-scale deletions or translocations induced by NHEJ inhibition can be reduced by adding a small molecule that blocks PolΘ, thereby inhibiting MMEJ (Supplementary Fig. 2). It is worth noting that the combination of M3814 and PolQi2 was more efficient for SNIPE than AZD7648 and PolQi2. However, AZD7648 was used at 1 μM and M3814 at 2 μM, according to previous publications. Increasing AZD7648's concentration from 1 μM to 2 μM might improve SNIPE performance to comparable levels. In line with a previous study[2], we show that these unintended events can be almost entirely prevented for some targets by instead adding an siRNA mix that targets *POLQ* mRNA. While no or few clones analyzed for several targets showed copy number changes, the *IZUMO4* target stood out with 40% of clones with copy number loss. When assessing the frequency of copy-number-neutral loss of heterozygosity that is indicative of extensive copying from the sister chromatid, three targets showed 0%, two targets 10–17%, and one target 100%. Because of this range and previous studies where we found 8–10% or 3% of cellular clones with copy-neutral loss of heterozygosity when using no pathway inhibition[82] or combined NHEJ and MMEJ inhibition[2], respectively, the tendency for copy-number-neutral loss of heterozygosity is likely to be target-site dependent rather than caused by pathway inhibition. This calls for careful evaluation of the aforementioned unintended side effects in each experiment.

We show that SNIPE can be used in conjunction with Cas9, HiFi-Cas9, Cas12a (Cpf1-Ultra), and ErCas12a (MAD7) and in human and chimpanzee iPSCs and several other cell lines (K562, RD-ES, HT-29, HuCC-T1). SNIPE does not only increase precise editing, but also improves outcome purity by additionally targeting editing-induced indels that do not destroy the gRNA recognition site (Supplementary Fig. 5). However, if the initial editing approach results in many indels that destroy the gRNA recognition site, SNIPE would enrich for such unwanted events as well. To avoid selection of unintended indel edits, SNIPE can presumably be used to enrich outcomes from DSB-free single-nick PE[35] and other editing methods such as DNA polymerase editors[83], and those relying on integrases or recombinases[5,84].

To allow SNIPE selection of cells based on changes in the transcriptome rather than the genome, one could combine SNIPE with the "raptor" method, in which a reprogrammed tracrRNA can bind endogenous cellular RNA to form a DNA-targeting CRISPR RNP[85]. If the edited position lies in a highly expressed gene, enzymes such as Cas12a2 could potentially be used to enrich edited cells by RNA-guided, collateral cleavage of adjacent DNA[86].

Inhibition of NHEJ and MMEJ during SNIPE is suitable for genome editing of cells, but not for genome editing in organisms, due to substantial cell death of unedited cells. However, SNIPE can be used not only in conjunction with genome editing but also to select against cells with genotypes that have not been introduced by editing. For example, it can kill cells that carry homozygous pathogenic cancer mutations (Fig. 5c). Intriguingly, it also increases the frequencies of wild type alleles in cancer lines carrying heterozygous mutations in the genes *TP53, APC* and *KRAS*, presumably by generating cells that lack the pathogenic mutation. This suggests that it induces repair of cancer mutations using the homologous chromosomes as templates. This mechanism could potentially be exploited for ex vivo repair of patient cells carrying heterozygous dominant cancer or other pathogenic mutations. For any potential treatment of tumors in vivo, the transient combined inhibition of NHEJ and MMEJ would obviously have to be tested for toxicity. Interestingly, mice treated with both a NHEJ inhibitor and a PARP1 inhibitor for 8 weeks showed no signs of distress during the course of the therapy and continued to remain well for at least four more months after the treatment[87]. PARP1 is not a bona-fide MMEJ pathway protein, but small molecule PARP1 inhibition has been shown to strongly decrease MMEJ-mediated deletions at DSBs[88]. To reduce reagent complexity and improve delivery potential, inhibition of MMEJ could also be achieved by a small molecule inhibitor of PolΘ[89] rather than a siRNA mix targeting *POLQ*. It is promising that small

molecule inhibitors of both the NHEJ and the MMEJ pathways are already in clinical trials for the treatment of cancers[90–93] and that in contrast to state-of-the-art cancer treatment using chemotherapy, which introduces DSBs in any dividing cell, SNIPE would selectively induce DSBs in cancer cells.

## Methods

### Cell culture

We used a 409B2 human induced pluripotent stem cell line (Riken BioResource Center, catalog no. HPS0076, GMO permit AZ 54-8452/26), as well as a variant carrying an iCRISPR-Cas9[94]. Furthermore, we modified iCRISPR 409B2 to carry a reverse transcriptase adjacent to Cas9H840A[35] (iPrime) as described in Riesenberg and Maricic 2018[94]. The chimpanzee stem cell line Sandra A was generated in a previous study by reprogramming of isolated leukocytes[95]. Cells were cultured at 37 °C and 5% CO2 in mTeSR1 medium (StemCell Technologies, catalog no. 05851) with supplement (StemCell Technologies, catalog no. 05852) on Matrigel Matrix (Corning, catalog no. 35248). Medium was changed daily. Cells were passaged 1:5 to 1:10 when they reached ~80% confluence by dissociation with EDTA (VWR, catalog no. 437012 C) for 6 min at room temperature and plated in medium containing 10 µM Rho-associated protein kinase (ROCK) inhibitor Y-27632 (Calbiochem, catalog no. 688000) for 24 h. For cryopreservation, detached cells were resuspended in mFresR (StemCell Technologies, catalog no. 5854), gradually frozen to –80 °C in an isopropanol box and stored in liquid nitrogen. For the generation of cellular clones, stem cells were treated with TrypLE (Gibco, catalog no. 12605010) for 5 min at 37 °C and single cells were sorted by a single cell sorter (Cytena Single-Cell Printer) into 96-well plates containing conditioned StemFlex medium (Gibco, catalog no. A3349401) supplemented with 10% CloneR (StemCell Technologies, catalog no. 05888). After 10–14 days, colonies could be passaged and genotyped.

Human colorectal adenocarcinoma cells HT-29 (Cytion, catalog no. 300215) were cultured in EMEM (Sigma-Aldrich, catalog no. M4655) supplemented with 10% FBS (Gibco, catalog no. 10270-106) and human CML cells K562 (ECACC, catalog no. 89121407) were cultured in IMDM (Gibco, catalog no. 12440053) with 10% FBS. Human intrahepatic cholangiocarcinoma cells HuCC-T1 (Cytion, catalog no. 300469), the human bone sarcoma cell line RD-ES (Cytion, catalog no. 300410) and the human leukemia cell line THP-1 (Cytion, catalog no. 300356) were cultured in RPMI 1640 (ThermoFisher, catalog no. 11875-093) with 10% FBS. Medium was refreshed every 2–3 days, and cells were passaged using Accutase (Sigma, catalog no. A6964) once per week in a 1:3 to 1:8 ratio. For cryopreservation, medium with 10% DMSO (Life Technologies, catalog no. D12345) was used.

All cell lines were tested negative for mycoplasma contamination and authenticated by the supplier via certificate of analysis as well as in-house morphology check.

### Oligonucleotides and small molecules

All gRNAs, DNA donors and primers were ordered from Integrated DNA Technologies (see Supplementary Data: List of oligonucleotides). Alt-R CRISPR-Cas9 tracrRNA was purchased as well from Integrated DNA Technologies (catalog no. 1072534). Small molecules M3814 (catalog no. HY-101570), AZD7648 (catalog no. HY-111783) and PolQi2 (catalog no. HY-150279) were ordered from MedChemExpress.

### Electroporation

409B2 iPrime cells were treated with 2 µg ml⁻¹ doxycycline (Clontech, catalog no. 631311) 3 days prior to editing to express the prime editor. For electroporation, single cell suspensions were obtained by treatment of adherent cells with TrypLE (Gibco, catalog no. 12605010) for 5 min at 37 °C and trituration. Preheated medium was added before cells were counted using the Countess Automated Cell Counter (Invitrogen). One million cells per electroporation were added into 5 ml

PBS (Gibco, catalog no. 14190250) and centrifuged at $200 \times g$ for 5 min at room temperature. For all cells except the 409B2 iPrime cell line, *Streptococcus pyogenes* Cas9-HiFi (R691A) (IDT, catalog no. 1081061), *Acidaminococcus* sp. BV3L6 Cas12a (Cpf1-Ultra) (IDT, catalog no. 10001273), or the *Streptococcus pyogenes* Cas9 adenine base editor ABE8e (Protein Production Sweden PPS) were used. Single cell suspensions were transferred into cuvettes containing 252 pmol CRISPR enzyme if applicable, 100 µl Human Stem Cell nucleofection buffer (Lonza, catalog no. VVPH-5022), 100 pmol electroporation enhancer (IDT, catalog no. 1075916 or 1076301), 320 pmol gRNA (crRNA/tracR duplex for Cas9 and crRNA for Cas12a) (or 640 pmol pegRNA and 214 pmol nicking gRNA for prime editing) and 200 pmol of single-stranded DNA donor or 10 µg of ds DNA donor if applicable. For edits with HDRobust, 320 pmol of *POLQ* siRNA - SMART 10721 and 640 pmol of *POLQ* siRNA 765 were added into the cuvette and 2 µM M3814 was added to the medium after nucleofection for 2 days. Where applicable, 3 µM of PolQi2 and 1 µM of AZD7648 were added[38]. For all cell types and edits, program B-16 of the Nucleofector 2b device (Lonza) was used for the electroporation and ROCK inhibitor Y-27632 was added for the first 24 h.

### Illumina library preparation and sequencing

At least 3 days after editing when cells reached confluency they were dissociated using EDTA (VWR, catalog no. 437012 C), pelleted and resuspended in 30–50 µl QuickExtract DNA extraction solution (Lucigen, catalog no. QE09050). Single-stranded DNA was extracted from the cells by incubation at 65 °C for 10 min, 68 °C for 5 min and 98 °C for 5 min. Target PCRs were performed in a total volume of 25 µl using the KAPA2G Robust PCR Kit (Sigma, catalog no. KK5024) with supplied buffer B and 3 µl of cell extract and incubated in a T100 Thermal Cycler (Bio-Rad) at 95 °C 3 min; 34× (95 °C 15 s, 65 °C 15 s, 72 °C 15 s); 72 °C 60 s. In a second PCR, Illumina adapters (P5 and P7) with sample-specific indices were added using Phusion HF MasterMix (Thermo Scientific, catalog no. F-531L) and 0.3 µl of the first PCR product. After cycling at 98 °C 30 s; 25× (98 °C 10 s, 58 °C, 10 s, 72 °C 20 s); 72 °C 5 min, amplicons were checked on a 2% EX agarose gel (Invitrogen, catalog no. G4010–11), pooled and purified using solid phase reversible immobilization beads in a 1:1 ratio of beads to PCR solution[96]. Libraries were sequenced as paired-end sequences of 2 × 151 bp (+7 bp index) on a MiSeq (Illumina) and base called with Bustard (Illumina). Adapter trimming was done using leeHom[97].

### Amplicon sequence analysis

After sequencing, demultiplexing and conversion of bam files into fastq files was done using SAMtools v.1.12[98]. Sequencing read percentage of targeted nucleotide substitution (HDR or PE for prime editing), indels (NHEJ and MMEJ) and mix of both (imperfect HDR or imperfect PE for prime editing) were calculated on the fastq files using CRISPResso1[99]. The analysis was restricted to amplicons with a minimum of 70% similarity to the wild type sequence and to a window of 20 bp from each gRNA. Substitutions were ignored to avoid false characterization of sequencing errors as NHEJ and sequence similarity for HDR occurrence was set to 95%.

### Resazurin assay

To measure viability, a resazurin assay was performed. The amount of living cells can be determined by fluorescent measurements as cellular dehydrogenases convert resazurin into fluorescent resofurin (excitation: 530–570 nm, emission: 590–620 nm). Cells were cultured for 3 days after editing, then fresh medium containing 10% resazurin solution (Cell Signaling, catalog no. 11884) was added. After 2 h, fluorescence was measured using a CLARIOstar imager (BMG Labtech). To detect background fluorescence, empty wells were filled with medium containing resazurin.

## PCR and gel electrophoresis

To genotype cellular clones of large knock-ins, the inserts were amplified with the KAPA2G Robust PCR Kit (Sigma, catalog no. KK5024) with supplied buffer B and 3 µl of cell extract. The cycling protocol for the PCR was: 95 °C 3 min; 34× (95 °C 15 s, 65 °C 15 s, 72 °C 5 min); 72 °C 60 s. Afterwards, the PCR products were analyzed in an automated parallel capillary electrophoresis using the Fragment Analyzer PROSize 2.0 (Agilent) with the dsDNA 920 Reagent Kit (Agilent, catalog no. DNF-920). The wild type cell line was taken along as a control. Successful knock-ins were defined if a peak with the expected size could be detected.

## Droplet digital PCR

A quantitative droplet digital PCR was performed to measure the copy number of single-cell-derived cellular colonies. Primers were designed to flank the cut site while the probe coupled to FAM was designed to exclude the edited site. *FOXP2* coupled to HEX was used as a reference gene. For the reaction, 1× ddPCR Supermix for probes (no dUTP, Bio-Rad, catalog no. 1863024) was mixed with 0.2 µM primers and 0.2 µM probe for target and reference and with 1 µl genomic DNA in QuickExtract DNA extraction solution (Lucigen, catalog no. QE090500). After droplet generation, the following cycling protocol was run: 5 min at 95 °C, 40 cycles 30 s at 95 °C (ramp rate 2 °C s$^{-1}$), 60 s at 61 °C for *DCHS1* and 58 °C for the puromycin resistance gene, *AHR, CDH16, IZUMO4, NCOA6* and *PROM2* (ramp rate 0.4 °C s$^{-1}$) and 5 min at 98 °C. Afterwards, droplets were analyzed in a QX200 Droplet reader (Bio-Rad). For Supplementary Fig. 2, the QIAcuity Digital PCR System (Qiagen) was used according to the manufacturer's instructions with the following cycling protocol: 2 min at 95 °C, 40 cycles 15 s at 95 °C and 30 s at 58 °C. Copy numbers were determined by the ratio of fluorescent signal between the target and the reference gene.

## Off-target analysis

To analyze off-targets of gRNAs used for SNIPE, CRISPOR[44] was used to predict the top two off-targets for every gRNA based on the CFD off-target score for Cas9 Hifi gRNAs. For Cpf1 gRNAs, due to the absence of a score, the top two candidates of the list of off-targets were chosen. Primers were designed for both off-targets per gRNA, and the loci were sequenced for all edited bulks following the Illumina library preparation and sequencing as well as analysis part of the methods section.

## Statistics and reproducibility

Bar graphs in figures were plotted and s.e.m. error bars were calculated using GraphPad Prism 10 software. The number of replicates is stated in the respective figure legends. No statistical method was used to predetermine sample size. The experiments were not randomized. Samples were prepared unblinded but in parallel. Analysis was performed on the basis of numerical sample names, without the identity of the samples being known during the analysis.

## Reporting summary

Further information on research design is available in the Nature Portfolio Reporting Summary linked to this article.

## Data availability

The sequencing data generated in this study are deposited in the NCBI's Sequence Read Archive (SRA) with the accession code PRJNA1357864. Source data are provided with this paper.

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

## Acknowledgements

We thank Antje Weihmann and Barbara Schellbach for DNA sequencing as well as Marvin Rößler and Anna Eccarius for DNA library preparation. We thank Dominik Macak for his help in the cell culture. We acknowledge Protein Production Sweden (PPS) for providing facilities and experimental support, (and we would like to thank Tomas Nyman and Henry Ampah-Korsah for assistance). PPS is funded by the Swedish Research Council as a national research infrastructure. Funding was provided by the Max Planck Society (S.P.), the NOMIS Foundation (S.P.), BMBF GoBio-Initial program (S.R.).

## Author contributions

S.R. conceived the idea. S.R. and L.F. designed the study. L.F., S.R., M.O., P.K., T.S., D.C. and E.R. performed experiments. P.K and M.O. performed the puromycin kill and selection experiment. S.R. tested SNIPE for single-nucleotide edits. P.K., S.R., T.S., D.C. and M.O. conducted the cancer cell experiments. L.F. and M.O. generated the iTF cell lines. T.S. generated the iMAD7 line. E.R. performed the off-target sequencing library. S.R., M.O. and L.F. generated the ancestral proteome cell line. L.F. and S.R. analyzed data. L.F., S.R. and S.P. wrote the paper with input from all authors.

## Funding

## Competing interests

Related patent applications on compounds for dual end-joining inhibition (patent applicant: Max Planck Society; inventors: S.R. and Tomislav Maricic; application number: EP18215071.4; status: pending), and improved gRNAs (patent applicant: Max Planck Society; inventors: S.R.,

Nelly Helmbrecht and Tomislav Maricic); application number: EP21176366.9; (status pending) have been filed. All other authors declare no competing interests.
