## [Transparent Peer Review file · Nature Communications]

Search-and-remove genome editing allows selection of cells by DNA sequence

Corresponding Author: Dr Stephan Riesenberger

Version 0:

Reviewer comments:

Reviewer #1

(Remarks to the Author)

The authors have addressed my main concerns.

Minor comment: The cell lines used in each figure and supp. figure should be stated in the legend.

Reviewer #2

(Remarks to the Author)

All my questions have been adequately addressed. The authors added new data/analysis. Specifically, SNIPE was tested in WT/WT (wild type), PE/PE, PE/WT, and PE/indel clones. Further, unintended target site copy number loss was analyzed.

Reviewer #4

(Remarks to the Author)

The authors' revised experiments provide additional data supporting the expanded use of SNIPE to enrich base-edited cells and dual-nuclease-based events.

Despite the new experiments, I still see inconsistencies in the study, as outlined below:

- 1) Regarding large genomic rearrangements, the authors main point is to confirm previously published data suggesting that dual inhibition of DNA-PKcs (M3814) and POLQ (siRNA) may decrease unintended copy number loss compared to DNA-PKcs inhibition alone. However this approach doesn't address concerns about more global DNA rearrangements within the entire edited cell population because the analysis is limited to selected single clones. For a complete assessment of editing safety and efficacy, it is essential to perform pooled, population-wide analyses using unbiased techniques such as CAST-Seq, as described by Corn and Cathomen's lab (PMID: 39604565), which more accurately quantify all genomic events after gene editing. Without such data, the conclusions reached by the clone-based analysis remain incomplete and inconclusive.
- 2) The comparisons presented between siRNA-mediated POLQ inhibition and chemical inhibition, as well as between DNA-PKcs inhibition using two different drugs at different concentrations (for instance 2 μ M for M3814 and 1 μ M for AZD7648), introduce significant bias into the analysis and lead to questionable conclusions. Without proper titration and direct assessment of target engagement for each inhibitor, it is not possible to accurately determine relative efficacy. Drawing conclusions about efficacy of treatment under these limited and non-standardized conditions is potentially misleading. Moreover, the materials and methods section provides insufficient information about compound usage. If the goal is to compare different treatments, the authors should undertake more careful validation, such as dose-response experiments and target engagement assays to ensure that comparisons between siRNA and small molecule inhibitors are meaningful.
- 3) The data presented focus on on-target alterations and do not address the issue of off-target editing, which remains a significant concern especially after two rounds of editing, even when using high-fidelity Cas9 variants. Off-target double-strand breaks (DSBs) can occur in both edited and non-edited cells, potentially reducing the overall experimental yield. This

issue is particularly relevant for primary cells, where a reliable purification or selection method is particularly needed to enrich for successfully edited cells. It is notable that the authors have not addressed the previous suggestion to perform experiments in primary cells, which would help demonstrate the applicability and robustness of their approach in more challenging cellular contexts, more sensitive to multiple DSBs. As pointed out above including unbiased genome-wide assays or predictive tools would strengthen confidence in the findings and help ensure that unintended edits are rigorously excluded and not actually enriched by this selection method.

Version 1:

Reviewer comments:

Reviewer #4

(Remarks to the Author)

I thank the authors for their clarifications although they don't fully address my points:

1) I believe the first response overstates what clone-based ddPCR and target amplicon NGS can tell you. These assays are locus-centric, subject to viability/expansion bias, and cannot provide a genome-wide view of balanced translocations, inversions, complex rearrangements, or aneuploidy. Equating "on-target clonal" analysis with "global rearrangements in a pool of cells" is misleading; pooled, unbiased methods would be important to characterize the full event spectrum and distal junction partners across the edited population. This will provide relevant information to the end users, helping them select this method over alternative options.

Saying "dozens of clones" are sufficient lacks statistical justification and the IZUMO4 example underscores target-dependent variability that weakens generalization. The dismissal of CAST-Seq focuses on its limitations (that have been well described in previous literature) but ignores its complementary strengths for pooled junction discovery; arguing it wouldn't add "meaningful data" is inconsistent also considering the authors don't provide any experimental proof or previous data from literature to justify this.

A constructive path is to acknowledge the complementary roles, add pooled, population-wide profiling (e.g., CAST-Seq for rearrangements or low-pass WGS for CNV/aneuploidy) at least on selected targets.

2) While aligning inhibitor concentrations with prior publications generally support comparability, the cited studies optimized dosing for HDR enhancement where cells can exploit alternative pathways (HDR) to promote cell survival rather than pathway suppression or cytotoxic endpoints, which may necessitate different exposure levels for the SNIPE context. The use of M3814 at 2 μ M versus AZD7648 at 1 μ M, without dose-response in cytotoxic context or target engagement measurements, makes efficacy comparisons across modalities potentially confounded by differential mechanism (HDR vs DSB dependent cell death). However a statement indicating these limitations is acceptable.

3) As mentioned in point 1, the authors' reply does not address the request for unbiased, genome-wide off-target assessment—especially relevant after two editing rounds in primary cells. Predictive, locus-limited assays can miss unanticipated off-target DSBs and complex rearrangements; pooled, unbiased methods would strengthen confidence that unintended edits are not enriched by SNIPE. A practical path to address the point about primary cells is to add an explicit limitations statement where the authors acknowledge that the work is optimized in cell lines amenable to clonal derivation and has not been validated in primary cells.

Dear Reviewers,

We would like to thank the initial three reviewers as well as the fourth additional reviewer for their valuable input on our manuscript. We are very grateful for the time you have invested in improving our manuscript. Please find below a point-by-point response to your comments.

Sincerely,

Dr. Stephan Riesenber

Department of Evolutionary Genetics

MPI for Evolutionary Anthropology

Deutscher Platz 6, D-04103 Leipzig, Germany

Point-by-point response

Reviewer #1:

The authors have addressed my main concerns.

Minor comment: The cell lines used in each figure and supp. figure should be stated in the legend.

We are happy to hear that and have now added information about the cell lines used to the legends.

Reviewer #2:

All my questions have been adequately addressed. The authors added new data/analysis. Specifically, SNIPE was tested in WT/WT (wild type), PE/PE, PE/WT, and PE/indel clones. Further, unintended target site copy number loss was analyzed.

We are glad to have adequately addressed your comments and questions. Thank you for the idea to test SNIPE on clones with different genotypes and for the appreciation of the copy number loss analysis we did.

Reviewer #4:

The authors' revised experiments provide additional data supporting the expanded use of SNIPE to enrich base-edited cells and dual-nuclease-based events.

Despite the new experiments, I still see inconsistencies in the study, as outlined below:

1) Regarding large genomic rearrangements, the authors main point is to confirm previously published data suggesting that dual inhibition of DNA-PKcs (M3814) and POLQ (siRNA) may decrease unintended copy number loss compared to DNA-PKcs inhibition alone. However this approach doesn't address concerns about more global DNA rearrangements within the entire edited cell population because the analysis is limited to selected single clones. For a complete assessment of editing safety and efficacy, it is essential to perform pooled, population-wide analyses using unbiased techniques such as CAST-Seq, as described by Corn and Cathomen's lab (PMID: 39604565), which more accurately quantify all genomic events after gene editing. Without such data, the conclusions reached by the clone-based analysis remain incomplete and inconclusive.

Indeed, we confirm that dual inhibition of DNA-PKcs and POLQ decreases unintended copy number loss compared to DNA-PKcs inhibition. We firmly believe that our approach of generating cellular clones and analysing copy number loss by ddPCR as well as target sequence by NGS is sufficient (and in some labs even considered gold-standard) to assess any on-target effect including global DNA rearrangements. It does accurately quantify the frequency of cells suffering from various kinds of unintended on-target effects. This combination detects insertions and deletions that can involve large chromosomal regions, gene conversion of one chromosome to match the sequence of the other chromosome, chromosomal rearrangements including translocations, inversions and aneuploidy (PMID: 36620901). Thus, our approach does not miss unintended on-target effects including rearrangements.

Even though the aim of many researchers is to obtain single clones after editing and selection, our analysis is importantly not limited to selected single clones. We report all clones from unbiased single-cell sorting, and mostly have data for dozens of clones. For the *FANCF* target and SNIPE method we used throughout the manuscript (M3814 + POLQ siRNA) we analysed 79 clones. Overall, we analysed 297 clones for the paper. A cell population is just the sum of many clones and any percentage of rearrangements seen in the entire cell population will be mimicked in a subset of it. Since we report on-target effects in the order of single- to double-digit percentages and not promille or less, data of dozens of clones is sufficient to describe a meaningful frequency of on-target effects. In the discussion we write: 'While no or few clones analyzed for several targets showed copy number changes, the IZUMO4 target stood out with 40% of clones with copy number loss.'

Consequently, doing additional CAST-seq which has a low detection limit for some types of rearrangements would not add significantly more meaningful data. More importantly, a review about the detection and prevalence of structural variants generated by gene-editing in human cells (PMID: 37093294) points out that CAST-seq is limited in that it cannot detect on-target effects where the target primer site has been ablated and that quantification requires ddPCR calibration. Large deletions that are the most common on-target effects can ablate the primer site and thus CAST-seq can be even blind for some types of frequent large deletions in a cell population.

2) The comparisons presented between siRNA-mediated POLQ inhibition and chemical inhibition, as well as between DNA-PKcs inhibition using two different drugs at different concentrations (for instance 2 μ M for M3814 and 1 μ M for AZD7648), introduce significant bias into the analysis and lead to questionable conclusions. Without proper titration and direct

assessment of target engagement for each inhibitor, it is not possible to accurately determine relative efficacy. Drawing conclusions about efficacy of treatment under these limited and non-standardized conditions is potentially misleading. Moreover, the materials and methods section provides insufficient information about compound usage. If the goal is to compare different treatments, the authors should undertake more careful validation, such as dose-response experiments and target engagement assays to ensure that comparisons between siRNA and small molecule inhibitors are meaningful.

The reviewers asked for inclusion of experiments to relate our data to the study from Corn lab using AZD7648 and investigating large-scale genomic alterations. Consequently, we used the same concentrations as they used in their paper (3 μ M of PolQi2 and 1 μ M of AZD7648) (PMID: 39604565). The same concentrations were chosen in an earlier paper from the Maresca lab, in which they titrated different drug concentrations (PMID: 37580318). An optimal concentration of M3814 had been previously titrated as well (PMID: 31392986). In our methods we write that: 'Small molecules M3814 (catalog no. HY-101570), AZD7648 (catalog no. HY-111783) and PolQi2 (catalog no. HY-150279) were ordered from MedChemExpress.[...]...2 μ M M3814 was added to the medium after nucleofection for two days. Where applicable, 3 μ M of PolQi2 and 1 μ M of AZD7648 were added'. Thus, we believe that our materials and methods section provide sufficient information about compound usage.

Finally, the goal was not to compare different small molecule treatments, but rather to compare the SNIPE method, which uses siRNAs, with published potential small molecule alternatives. Opting for 2 μ M AZD7648 instead of 1 μ M used in prior publications would have compromised the comparability to previous studies. To avoid misleading conclusions we have now added following sentence to the discussion: 'It is worth noting that the combination of M3814 and PolQi2 was more efficient for SNIPE than AZD7648 and PolQi2. However, AZD7648 was used at 1 μ M and M3814 at 2 μ M, according to previous publications. Increasing AZD7648's concentration from 1 μ M to 2 μ M might improve SNIPE performance to comparable levels.'

3) The data presented focus on on-target alterations and do not address the issue of off-target editing, which remains a significant concern especially after two rounds of editing, even when using high-fidelity Cas9 variants. Off-target double-strand breaks (DSBs) can occur in both edited and non-edited cells, potentially reducing the overall experimental yield. This issue is particularly relevant for primary cells, where a reliable purification or selection method is particularly needed to enrich for successfully edited cells. It is notable that the authors have not addressed the previous suggestion to perform experiments in primary cells, which would help demonstrate the applicability and robustness of their approach in more challenging cellular contexts, more sensitive to multiple DSBs. As pointed out above including unbiased genome-wide assays or predictive tools would strengthen confidence in the findings and help ensure that unintended edits are rigorously excluded and not actually enriched by this selection method.

We do address off-target editing. Supplementary Fig. 7 and 8 are dedicated to analysis of the top two predicted off-targets for each target edit done in this study.

We write in the results: 'To assess off-target editing we scored the editing efficiency of the two most likely predicted off-target sites for each of the targets for which Cas9 and Cas12a were used in this study (PMID: 26780180, PMID: 29762716). Mean off-target editing with standard editing or SNIPE was 2.7% (0-47.2%) or 0.2% (0-2.1%), respectively. The CDH16 edit, which stood out with 47.2% combined off-targets with standard editing, was reduced to 2.1% with

SNIFE (22-fold) (Supplementary Fig. 7 and 8).’ We also have a corresponding ‘Off-target analysis’ section in the Methods.

We agree that it is always nice to test methods in several different cell types. We did not perform SNIFE in primary cells because we focused our efforts on the comprehensive set of other experiments central to the requests of the reviewers and have already tested SNIFE in human and chimpanzee iPSCs and several other cell lines (K562, RDES, HT29, HuCCT1), which we believe sufficiently validates our findings.

Nevertheless, in our previous response we acknowledged potential limitations and mentioned alternatives for use in e.g. primary cells as suggested by reviewer 3: ‘SNIFE requires the highly efficient delivery of potent ribonucleoproteins, which can be achieved for most common cell lines, including stem cells. However, CRISPR components are sometimes delivered encoded in plasmids that can be modified easily beforehand, but the delivery of plasmids is typically less efficient. For approaches with inefficient delivery, SNIFE should be combined with other methods developed to enrich for cells with successfully delivered CRISPR components (PMID: 33903218, 32916603) or cleavage activity on a plasmid surrogate reporter (PMID: 21983922). Also, the sequential delivery process may be less tolerated by certain cell types like primary cells, for which SEED (PMID: 39910194) or SLEEK (PMID: 37127662) selection might be suitable alternatives, as they can achieve efficiencies of more than 90%.’

Dear Reviewers,

Please find below a point-by-point response to your comments.

Sincerely,

Dr. Stephan Riesenber

Department of Evolutionary Genetics

MPI for Evolutionary Anthropology

Deutscher Platz 6, D-04103 Leipzig, Germany

Point-by-point response

Reviewer #4

I thank the authors for their clarifications although they don't fully address my points:

1) I believe the first response overstates what clone-based ddPCR and target amplicon NGS can tell you. These assays are locus-centric, subject to viability/expansion bias, and cannot provide a genome-wide view of balanced translocations, inversions, complex rearrangements, or aneuploidy. Equating "on-target clonal" analysis with "global rearrangements in a pool of cells" is misleading; pooled, unbiased methods would be important to characterize the full event spectrum and distal junction partners across the edited population. This will provide relevant information to the end users, helping them select this method over alternative options.

Saying "dozens of clones" are sufficient lacks statistical justification and the IZUMO4 example underscores target-dependent variability that weakens generalization. The dismissal of CAST-Seq focuses on its limitations (that have been well described in previous literature) but ignores its complementary strengths for pooled junction discovery; arguing it wouldn't add "meaningful data" is inconsistent also considering the authors don't provide any experimental proof or previous data from literature to justify this.

A constructive path is to acknowledge the complementary roles, add pooled, population-wide profiling (e.g., CAST-Seq for rearrangements or low-pass WGS for CNV/aneuploidy) at least on selected targets.

We thank the reviewer for the clarification of their point. To address these limitations, we toned down our previous statement about on/off target specificity and added the following paragraph in the discussion:

‘Our data suggest that SNIPE can also prevent unintended on-target editing events when clones are analyzed, as well as off-target editing events at predicted sites (Supplementary Fig. 1, 2, 6, 8). To improve quality control and increase sensitivity in the detection of unintended side effects in pooled cell populations, profiling using CAST-seq (PMID: 33626327) or low-pass whole genome sequencing for copy number variants and aneuploidy could be employed. Additionally, unbiased genome-wide off-target assessment (PMID: 28459458, 31000663) is necessary to exclude off-target editing events more rigorously.’

2) While aligning inhibitor concentrations with prior publications generally support comparability, the cited studies optimized dosing for HDR enhancement where cells can exploit alternative pathways (HDR) to promote cell survival rather than pathway suppression or cytotoxic endpoints, which may necessitate different exposure levels for the SNIPE context. The use of M3814 at 2 μ M versus AZD7648 at 1 μ M, without dose–response in cytotoxic context or target engagement measurements, makes efficacy comparisons across modalities potentially confounded by differential mechanism (HDR vs DSB dependent cell death). However a statement indicating these limitations is acceptable.

We agree with the argument made by the reviewer, but also believe that our statement indicating these limitations is sufficient.

3) As mentioned in point 1, the authors’ reply does not address the request for unbiased, genome-wide off-target assessment—especially relevant after two editing rounds in primary cells. Predictive, locus-limited assays can miss unanticipated off-target DSBs and complex rearrangements; pooled, unbiased methods would strengthen confidence that unintended edits are not enriched by SNIPE. A practical path to address the point about primary cells is to add an explicit limitations statement where the authors acknowledge that the work is optimized in cell lines amenable to clonal derivation and has not been validated in primary cells.

While we address the point on unbiased off-target assessment in our response to 1), we added an additional sentence in the discussion to acknowledge the limitations of SNIPE in primary cells.

‘SNIPE is optimized in cell lines amenable to clonal derivation, but has not been validated in primary cells.’